# Measuring Satisfaction and Comfort with Gender Identity and Gender Expression among Transgender Women: Development and Validation of the Psychological Gender Affirmation Scale

**DOI:** 10.3390/ijerph18063298

**Published:** 2021-03-23

**Authors:** Jae M. Sevelius, Deepalika Chakravarty, Samantha E. Dilworth, Greg Rebchook, Torsten B. Neilands

**Affiliations:** 1Center for AIDS Prevention Studies, Department of Medicine, University of California, San Francisco, CA 94158, USA; deepalika.chakravarty@ucsf.edu (D.C.); samantha.dilworth@ucsf.edu (S.E.D.); greg.rebchook@ucsf.edu (G.R.); torsten.neilands@ucsf.edu (T.B.N.); 2Center of Excellence for Transgender Health, Department of Medicine, University of California, San Francisco, CA 94158, USA

**Keywords:** transgender, gender affirmation, scale development, scale validation, mental health, substance use, gender identity, gender expression

## Abstract

Among transgender and gender diverse people, psychological gender affirmation is an internal sense of valuing oneself as a transgender or gender diverse person, being comfortable with one’s gender identity, and feeling satisfied with one’s body and gender expression. Gender affirmation can reduce gender dysphoria and mitigate deleterious health effects of marginalization. We sought to create an instrument to measure psychological gender affirmation among transgender women. Following initial item development using qualitative interviews, we used self-administered survey data from two distinct samples (*N1* = 278; *N2* = 368) of transgender women living with HIV in the USA. We used data from Study 1 to perform exploratory factor analysis (EFA) and data from Study 2 to perform confirmatory factor analysis (CFA), yielding the five-item single-factor Psychological Gender Affirmation (PGA) scale with high reliability (α = 0.88). This scale is psychometrically sound as demonstrated by its convergent and discriminant validity via correlations with select measures and by its predictive validity through associations in hypothesized directions with measures of mental health and substance use. The PGA scale will aid research on psychological gender affirmation that can in turn inform interventions as well as gender-affirming clinical and social practices to promote the health and well-being of transgender and gender diverse people.

## 1. Introduction

Gender identity is an internal felt sense of one’s gender, while gender expression is how a person presents their gender to the external world. Transgender and gender diverse people have a gender identity and/or expression that differs from the sex they were assigned at birth, whereas cisgender (i.e., non-transgender) people’s gender identity and expression align with their assigned birth sex. Further, some transgender people experience gender dysphoria, defined as psychological distress associated with a sense of incongruence between their gender identity and gender expression and/or body and physical appearance. Gender dysphoria is compounded by transphobia—the systemic stigma and discrimination faced by transgender people—which often results in social and economic marginalization as well as violence [1,2]. Experiences of transphobia are also significantly associated with poorer mental health and substance use among transgender and gender diverse people [1,3,4].

As conceptualized by the Model of Gender Affirmation [5], gender affirmation, whether social, medical, or psychological, has been demonstrated to reduce gender dysphoria and can mitigate some of the downstream negative health effects of marginalization [4,6,7,8,9,10,11]. Social gender affirmation is interpersonal recognition and respect for one’s gender identity, while medical gender affirmation entails the utilization of gender-affirming medical treatments, such as hormone therapy and/or surgeries. While social gender affirmation is important for people of all genders, not all transgender and gender diverse people pursue medical gender affirmation [8]. Building on the concept of self-affirmation from social psychology [12], psychological gender affirmation is an internal sense of valuing oneself as a transgender or gender diverse person, being comfortable with one’s own gender identity, and having a sense of satisfaction with one’s body and gender expression [10,13]. While all three types of gender affirmation (social, medical, and psychological) have been shown to improve health outcomes, the majority of research to date has focused on social and medical gender affirmation [1,8,14].

Although empirical research on psychological gender affirmation is scant, it is critical to the health and well-being of transgender and gender diverse people [10]. When psychological gender affirmation is low, discomfort can be manifested as not feeling comfortable going out during the day for fear of being mistreated by others, discomfort with being “out” as transgender and wishing to be perceived by others as cisgender [15]. These feelings contribute to social anxiety and even agoraphobia, which are more prevalent among transgender and gender diverse people [16]. A related yet distinct concept from psychological gender affirmation is internalized transphobia, defined as the internalization of transphobic societal attitudes [17]. Internalized transphobia is the negative appraisal of self and other transgender people due to an acceptance of rigid and binary social gender norms, while psychological gender affirmation describes the positive and potentially stress-buffering aspects of comfort and satisfaction with one’s gender identity and expression, without regard to congruence or conformity to society’s gendered expectations and judgments. Psychological gender affirmation is therefore a distinct and broader concept from internalized transphobia, and also likely serves as a protective factor against the internalization of transphobia.

Psychological gender affirmation is also a distinct concept from gender congruence, in that gender congruence indicates whether a person feels that their external appearance, or gender expression, is aligned with their gender identity [13], whereas psychological gender affirmation represents a felt sense of comfort and satisfaction with one’s identity and appearance, regardless of perceived alignment of one with the other. While congruence can be important to the health and well-being of transgender and gender diverse people, this congruence is not always possible or essential to a person’s sense of comfort and satisfaction with their gender identity and expression. For example, a person may identify as a woman, perceive her gender expression to be incongruent with her gender identity, and still be very comfortable and satisfied with her gender expression and identity. Further, gender congruence is largely dependent on culture-based and subjective gender norms regarding gender expression (e.g., cultural norms regarding what someone who identifies as a woman is “supposed” to look like). We were interested instead, in studying transgender women’s psychological gender affirmation, in terms of being observed by others (out in public during the day, where she may be identified as transgender), known by others as transgender (comfort with a transgender identity) and satisfaction with her own gender expression, body, and appearance (regardless of congruence with cultural norms based on gender identity).

While measures exist to assess internalized transphobia [17] and gender congruence [13], the lack of psychometrically sound measures of psychological gender affirmation is a barrier to further exploration of the construct. Transgender women experience extreme mental health and substance use disparities, with research consistently confirming disproportionately high rates of depression, anxiety, and substance use [18]. Thus, in alignment with the Model of Gender Affirmation, we sought to develop a measure of psychological gender affirmation within the context of mental health and substance use related health outcomes.

Using secondary data from two large quantitative studies of transgender women, the purposes of the current paper are to (a) describe the development of the Psychological Gender Affirmation (PGA) scale—a brief, self-report measure of comfort and satisfaction with one’s gender identity and expression, and (b) present evidence of the PGA scale’s relationship to hypothesized correlates. These hypothesized correlates were informed by our theoretical framework, the Model of Gender Affirmation, as well as previous research, and included variables related to hormone use, mental health, and substance use.

## 2. Methods

### 2.1. Item Development

We conducted individual qualitative interviews with 22 transgender women of color to explore constructs and associations proposed by the Model of Gender Affirmation and to develop potential items for this scale. Description of the recruitment methodology for that study has been documented elsewhere [5]. We used Atlas.ti [19] to analyze interview transcripts employing template analysis, which entails the development and application of a coding template for identifying and organizing themes in qualitative data [20]. Template analysis is useful when some a priori themes are informed by theory and/or the research questions of interest and are therefore pre-determined. For this study, the theoretical framework of the Model of Gender Affirmation informed the a priori themes. For the current study, the initial analysis phase included identification of thematic categories (e.g., “comfort with gender identity”) that were relevant to item development. The themes were then further catalogued into codes and modified as needed to ensure that the coding template included language used by the participants, thus generating the initial list of candidate items.

We used an iterative process involving two rounds of cognitive interviewing to test the initial items. Cognitive interviewing improves survey development through administration of draft items while obtaining further information from participants about their responses to the items [21]. During the interviews, we asked participants to respond to each candidate item for the scale. Following their response, we asked them to describe what they understood the item to be asking and to provide suggestions for improving the item, including appropriateness, wording, and/or the ordering of items in the scale. The initial items were slightly modified based on feedback from participants following the first round of cognitive interviews (*N* = 10). We then used the same technique to further refine the items in a second round of cognitive interviews with a different set of participants (*N* = 9). The five resulting items were included in surveys administered as part of two larger quantitative studies, described below.

### 2.2. Samples for the Current Analyses

We used data from two independent and distinct studies of transgender women living with HIV (hereafter referred to as Study 1 and Study 2) for the current analyses.

**Study 1—**Study 1 was designed to test the efficacy of “Healthy Divas”, an intervention for transgender women living with HIV to improve engagement in HIV care. The intervention was evaluated using a two-site randomized controlled trial conducted in two cities (San Francisco and Los Angeles) in the western United States of America. The study, which was conducted in accordance with the Declaration of Helsinki, was approved by the Western Institutional Review Board (20181370), and the Institutional Review Board at the University of California, San Francisco (15-17910). Self-administered behavioral surveys were conducted with participants (*N* = 278) at baseline and follow up visits. The baseline data was utilized for the current analysis.

**Study 2—**Participants were recruited at nine study sites for an initiative titled Enhancing Engagement and Retention in Quality HIV Care for Transgender Women of Color. The sites were in the West (San Francisco Bay Area and Los Angeles), Midwest (Chicago), and East (New York City) regions of the United States of America. The goal of the initiative was to develop and implement novel interventions to engage and retain in HIV care transgender women of color living with HIV [13]. A tenth study site was tasked with evaluating these interventions. The study was conducted in accordance with the Declaration of Helsinki and was approved by the local Institutional Review Board at each site. Specifically, the parent study which yielded the data for this analysis was approved by the Institutional Review Board at the University of California, San Francisco (12-09622). The current analysis utilizes data from the self-administered survey that participants completed at baseline. Many of the participants from the sites in the West region were common to both Study 1 and Study 2 because of the small size and tight-knit nature of the communities of transgender women. To ensure that the samples for the current analyses were independent and distinct, we excluded from our analyses the 486 participants from the Study 2 sample who were recruited at the study sites in the West region. Further, we excluded participants (*N* = 4) who did not respond to any of the candidate items in the scale being developed. These two exclusions yielded a final sample of 368 transgender women from Study 2 (Appendix A).

### 2.3. Screening and Recruitment

**Study 1—**Participants were recruited from November 2016 to October 2019 from community-based organizations, social networks, and local venues frequented by transgender women. To be eligible, one had to be at least 18 years of age; be assigned male sex at birth but identify as transgender female, female, or another transfeminine identity; confirmed to be living with HIV via medical documentation or an HIV rapid test; and be fluent in English or Spanish. After eligible participants provided informed consent to be enrolled in the study, baseline data was collected via a self-administered survey in CASIC [22] that included topics such as participants’ mental health, substance use, trauma history, and gender affirmation. Participants received $40 to compensate for their time at the baseline study visit.

**Study 2**—Participants were recruited from December 2013 to August 2016 using multiple strategies, including community outreach, publicity materials, and referrals from clinics and service providers. To be eligible, one had to be at least 16 years old, be assigned male sex at birth but identify as transgender or female, be living with HIV, and be fluent in English or Spanish. Participants visited the study offices to provide informed consent, be enrolled in the study and complete a self-administered survey in REDCap [23] that included topics such as their physical and mental health, substance use and gender affirmation. As compensation for their time, they received incentives between $25 and $50 depending on the study site.

### 2.4. Measures

*Candidate items for the new PGA scale:* Both studies’ surveys contained these five questions: “How comfortable are you with going out in public during the day?”, “How comfortable are you with people knowing that you are transgender?”, ‘How satisfied are you with your body the way it is right now?”, ‘How satisfied are you with the way you look right now?”, and ‘How satisfied are you with your current level of femininity?” Response options were 5-point Likert type: Not at all comfortable, Slightly comfortable, Moderately comfortable, Very comfortable, and Extremely comfortable for the first two items; Not at all satisfied, Slightly satisfied, Moderately satisfied, Very satisfied, and Extremely satisfied for the last three items.

*Sample characteristics*: The following self-reported data were used to characterize the two samples: age, level of education, race-ethnicity, sex work as a source of income in the previous six months, and experiences of homelessness in the previous six months.

*Financial security*: In Study 1, we asked the question ‘Which of the following statements best describes your financial situation?” with responses “I have enough money to live comfortably”, “I can barely get by on the money I have”, and “I cannot get by on the money I have”. Participants reporting the first response were categorized as financially secure. In Study 2, we asked the question “In the last 6 months, how many of those months did you run out of money for your basic necessities?” Participants who responded zero were categorized as financially secure and those whose response was 1 month through 6 months were categorized as financially insecure.

*Feminizing hormone therapy:* Participants reported if they had ever taken feminizing hormones (0 = No, 1 = Yes) and also if they were currently taking them (0 = No, 1 = Yes).

Additionally, data on the following study-specific measures from the two studies were used in the analyses:

Study 1: *Resilience,* ‘the ability to bounce back’, was measured using a 3-item adaptation of the Brief Resilience Scale [24] with response choices ranging from 1 (Strongly disagree) to 5 (Strongly agree). The sum of the items represented the scale score; Cronbach’s alpha was 0.75. Sample item: “It does not take me long to recover from a stressful event.”

*Transgender Group Identification,* adapted from the Multigroup Ethnic Identity Measure [25], consisted of 7 items with response choices that ranged from 1 (Strongly disagree) to 5 (Strongly agree). The mean of all items was calculated to represent a total score; Cronbach’s alpha was 0.82. Sample item: “I am active in organizations or social groups that include mostly transgender people.”

*Time since HIV diagnosis* was defined as the difference of the self-reported date of HIV diagnosis from the date of the baseline assessment, measured in years.

*Experiences of traumatic events:* Participants were asked if they had experienced a list of traumatic events from three categories adapted from the Trauma History Questionnaire [26]: being victim of a crime (e.g., robbery, mugging), being victim of physical or sexual violence (e.g., beating, rape), or experiencing general traumatic events (e.g., flood, fire, car accident). The sum of endorsed events from the 14 events in the general trauma category was used.

*Positive and negative affect*: To measure these we used the 10-item International Positive and Negative Affect Schedule Short Form (I-PANAS-SF) [27]; each of the two subscales contained 5 items. Participants rated the frequency of each emotion using five unipolar response choices ranging from 1 (Never) to 5 (Always). The sum of the five items was calculated for each subscale; Cronbach’s alphas were 0.88 and 0.85 for positive and negative affect, respectively. Sample item for negative affect: “Thinking about yourself and how you normally feel, to what extent do you generally feel upset?”

*Severity of alcohol use* was measured among those reporting alcohol use in the past 12 months by summing the yes responses to 15 yes/no items about their alcohol use, adapted from the DSM-V [28]. The summary score was then binned into 4 categories of severity: 0 (0–1, None), 1 (2–3, Mild), 2 (4–5, Moderate), and 3 (≥6, Severe). If no alcohol use was reported, the severity was calculated as 0. Sample item (reverse scored): “Have you tried to cut down or stop using alcohol?”

*Severity of illicit drug use* was defined in a similar fashion as for alcohol, above. The same 15 items used for alcohol were asked for each of 13 illicit substances (e.g., methamphetamine, marijuana, cocaine), with the word “alcohol” replaced with the name of the appropriate illicit substance. The total number of yes responses were calculated for each substance separately and the maximum of these 13 sums was used to determine the category of overall drug use severity: 0 (0–1, None), 1 (2–3, Mild), 2 (4–5, Moderate), and 3 (≥6, Severe). If no illicit drug use was reported, the severity was 0. Sample item: “Have you had withdrawal symptoms when you cut down or stopped using methamphetamine?”

Study 2:

*Body Satisfaction* was measured using a 7-item adaptation of the Body Appreciation scale [29]. The 5-point Likert type response options ranged from 1 (Strongly disagree) to 5 (Strongly agree). The mean of the items represented the scale score. Sample item: “On the whole, I am satisfied with my body.” Cronbach’s alpha for this scale was 0.94.

*Lifetime experiences of transphobia* were recorded using a 5-item measure [3] adapted from the Schedule of Racist Events [30]. The Likert type response options ranged from 1 (Never) to 6 (All of the time). The scale scores were dichotomized for use in predictive validity analyses (0: Never experienced transphobia; 1: Experienced transphobia in the past). Sample item: “How many times in your entire life have you been hit, shoved, or beaten up because you are transgender?” Cronbach’s alpha for the interval-level score was 0.77.

*Depression:* The 10-item Center for Epidemiological Studies Depression (CES-D) short scale was used to record depression in the previous week [31,32]. The response options are Likert type ranging from 0 (Rarely or none of the time (less than 1 day)) to 3 (Most or all of the time (5–7 days)). The scale score was calculated by reverse coding two positively-valenced items and then calculating the mean of all items. These composite scores were dichotomized for use in the predictive validity assessment model (0: composite score less than 10 indicating the absence of depression; 1: composite score of 10 or greater indicating the presence of depression) [33]. Sample item: “During the past week, I felt that everything I did was an effort.” Cronbach’s alpha for the interval-level depression score was 0.87.

### 2.5. Data Analyses

First, we computed the descriptive statistics—proportions, means and standard deviations—for both study samples using SAS software version 9.4 (SAS Institute, Inc., Cary, NC, USA) [34].

*Exploratory Factor Analysis (EFA)*: Second, we performed an EFA using the data from Study 1 on the five potential items for the new scale. To determine the optimal number of factors in the initial screening stage, we utilized the Hull method [35], available in the factor analysis software program FACTOR 10 [36], along with a scree plot produced from the eigenvalues. Items were retained in the final EFA model if they both followed the underlying theory of the construct as well as had acceptable factor loadings.

*Confirmatory Factor Analysis (CFA)*: With the factor structure from the EFA from Study 1, we performed the CFA using data from Study 2. Some participants in the sample (8.7%; *N* = 32) had missing data on a subset of the five PGA candidate items. To account for this, we used multiple imputation within Mplus (250 imputations) [37]. The chi-square test of exact fit was used to assess the global model fit. The approximate fit was assessed using the criteria that at least two of these three conditions are satisfied: Root Mean Square Error of Approximation (RMSEA) ≤ 0.06, Bentler’s Comparative Fit Index (CFI) ≥ 0.95, and the Standardized Root Mean Square Residual (SRMR) ≤ 0.08 [38]. Mplus version 8.4 [39] was used for the CFA. As a measure of the internal consistency reliability, we calculated Cronbach’s alpha [40] as well as composite reliability [41]; values of 0.7 and greater are considered acceptable for both these measures. We also calculated average variance extracted (AVE) for which values above 0.5 are considered acceptable and those above 0.7 are considered very good [42].

*Convergent and discriminant validity*: We used the correlations of the PGA composite score with composite scores of measures of interest from the two studies to evaluate these. We hypothesized that the PGA score would be positively correlated with resilience, transgender group identification, being on feminizing hormone therapy (ever and currently), and body satisfaction, thereby demonstrating convergent validity. We hypothesized that the PGA scale would demonstrate discriminant validity through the absence of statistically significant correlations with time since HIV diagnosis, experiences of traumatic events, experiences of transphobia, education level (dichotomized into less than college level vs. some college or higher), and financial security. These correlation analyses were conducted using full information maximum likelihood (FIML) in Mplus version 8.4 (Muthén & Muthén, Los Angeles, CA, USA).

*Predictive validity:* We evaluated this by examining the bivariate association of the PGA scale score with the following outcome variables: positive affect, negative affect, alcohol abuse, illicit drug use and depression. For the continuous outcomes of positive and negative affect, we performed ordinary least squares regression. The remainder of the outcome variables were binary and we performed logistic regression on them and obtained the odds ratio per unit change in the score of the PGA scale. We hypothesized that the PGA scale would be positively associated with positive affect, inversely associated with negative affect, and that higher PGA scale scores would be associated with lower odds of substance use and depression. All predictive validity analyses were conducted in SAS software version 9.4.

## 3. Results

*Participant characteristics: *Table 1 presents the descriptive characteristics of the participants in both studies. The racial and ethnic composition of the study participants were similar, as most participants in both studies were either Latina or non-Hispanic Black, with Study 1 including more multiracial, non-Hispanic participants (10.8% in Study 1 and 1.4% in Study 2). Levels of education were also similar across the two studies with the majority of participants reported education at grade 12 or lower. Participants in Study 1 were slightly older and fewer of them were financially secure. About two-thirds of the participants in Study 1 reported currently taking hormones (67.3%), as compared to just under half of the participants in Study 2 (47.0%). These sample characteristics per US geographic region appear in Appendix A.

*Exploratory Factor Analysis of Study 1 data:* The scree plot of eigenvalues sharply declined below 1.0 in between one and two factors and leveled off at three factors, indicating strong evidence for only one factor using the five items. The output from FACTOR’s Hull method indicated that a single common factor was advised. Loadings for all items onto one factor were acceptable (Table 2) and, since all five items remained conceptually practical, all were retained in the final one-factor solution. There was no missing data in this analysis.

*Confirmatory Factor Analysis of Study 2 data:* We used 250 imputed datasets to perform a CFA of the single 5-item factor indicated by the EFA. The null hypothesis of exact model-data fit was rejected (χ^2^(5) = 178.286, *p* < 0.001) but the approximate model-data fit was satisfactory with two of three fit statistics in the desirable bounds (RMSEA = 0.307, CFI = 0.980, SRMR = 0.052). The CFA factor loadings are presented in Table 2. Three of the five items demonstrated high loadings (>0.80) while two of them demonstrated moderately high loadings (>0.70). The reliability for the resulting five-item scale was high (Cronbach’s alpha = 0.88, composite reliability = 0.93). The average variance extracted (AVE) was 0.73. The final PGA scale is presented in the appendix.

*Convergent and discriminant validity using Study 1 and Study 2 data:* The PGA scale demonstrated statistically significant (*p* < 0.05) positive correlations with resilience, transgender group identification, being on feminizing hormone therapy—both ever and currently, and body satisfaction (Table 3). The PGA scale showed no significant correlation with time since HIV diagnosis, past experience of trauma, lifetime experience of transphobia, education level, and financial insecurity. All these results were as hypothesized.

*Predictive validity using Study 1 and Study 2 data:* The PGA scale demonstrated associations with select mental health and substance use outcomes in the expected directions (Table 4). Specifically, higher scores on PGA were associated with greater positive affect (β = 1.256, *p* = 0.001) and lower negative affect (β= –0.913, *p* = 0.002). Further, higher scores on PGA were associated with lower odds of alcohol abuse and dependence (OR = 0.662, *p* = 0.003), illicit drug abuse and dependence (OR = 0.737, *p* = 0.015) and depression (OR = 0.729, *p* = 0.003).

## 4. Discussion

The results of the factor analyses and the validity analyses suggest that a single-factor structure of the Psychological Gender Affirmation (PGA) scale fit the data well in the two research samples, and that comfort and satisfaction with gender identity and expression can be measured with a parsimonious five-item measure (see Appendix B for the final scale). The two comfort items reflect comfort with going out in public during the day and with people knowing about one’s transgender status or identity. The three satisfaction items reflect an internal sense of satisfaction with one’s body, appearance, and level of femininity regardless of perceived congruence with one’s identity. This distinction is important because we aimed to develop a measure of psychological gender affirmation that emphasizes one’s internal felt sense of comfort and satisfaction, which we propose may be even more feasible, empowering, and important to mental health than perceived congruence between one’s appearance and identity.

While medical gender affirmation often increases a sense of congruence for transgender people who choose to pursue it and have access to it, some transgender and gender diverse people do not have access to the full range of options, and some choose not to pursue medical gender affirmation. For some transgender and gender diverse people (and even cisgender people), complete congruence may not be possible due to genetics or other factors, especially given the unattainable ideals of gender presentation that are often held up by society as norms and then internalized. It is critical that transgender and gender diverse people who seek medical gender affirmation have access to it, as this type of affirmation is strongly associated with improved health outcomes. However, it is also critical for people of all genders to be able to feel comfortable and satisfied with one’s identity, body, and appearance, regardless of whether they conform to social norms regarding gender expression.

As expected, scores on the PGA scale were positively associated with use of feminizing hormones (ever and current), body satisfaction, resilience, and transgender group identification. As predicted by the Model of Gender Affirmation, psychological gender affirmation as measured by the PGA scale is correlated with medical gender affirmation as measured by current and past hormone use. For transgender people who use them, hormones can increase one’s sense of comfort and satisfaction with one’s identity and appearance. However, not all transgender women who are comfortable and satisfied with their gender identity and expression use hormones and vice versa. This variability is also demonstrated by the statistically significant yet low positive correlation between PGA and hormone use. Therefore, PGA and medical gender affirmation are related but distinct constructs. Similarly, scores on the PGA scale were associated with body satisfaction, a construct that is included within the scope of psychological gender affirmation. Resilience was positively associated with PGA scores, suggesting that comfort and satisfaction with one’s gender identity and expression may be a protective factor against the downstream negative health effects of transphobia-related stress. PGA scores were also associated positively with transgender group identification, consistent with literature that has demonstrated associations between identity pride, transgender community belongingness, and well-being [43,44].

Scores on the PGA scale were not associated with time since HIV diagnosis, education level, and financial security. Further, consistent with literature on internalized transphobia and minority stress, scores on the PGA scale were not associated with experiences of trauma or transphobia [17,45]. This finding supports the notion that psychological gender affirmation is an internal process that is distinct from experiences of victimization and/or transphobia from external sources. The lack of association between PGA scores and these measures contribute to the establishment of discriminant validity of the PGA scale, since these variables were not predicted to be associated with psychological gender affirmation.

Finally, the PGA scale demonstrated important associations with select mental and behavioral health outcomes in hypothesized directions. Greater psychological gender affirmation was associated with more positive affect, less negative affect, and lower depression. The construct of psychological gender affirmation includes elements of identity pride, as well as comfort and satisfaction with being out as transgender, and satisfaction with one’s gender expression and appearance. It follows that these elements of self-esteem would be associated with more positive affect and less negative affect and depression. It is also important that higher scores on PGA are associated with lower levels of use and dependence on alcohol and illicit drugs. Increasing psychological gender affirmation among transgender and gender diverse people may be a critical intervention point for reducing the extreme health disparities experienced by these communities in terms of mental health and substance use. The PGA scale may be useful in clinical and research settings to assess psychological gender affirmation alongside other forms of gender affirmation to inform interventions aimed at promoting the health and well-being of transgender and gender diverse people.

## 5. Strengths and Limitations

The samples used to develop and validate the PGA scale were community- and clinic-based and thus were not probability samples. Therefore, the generalizability of our results is limited. Further, limited variability of age, race, and ethnicity and relatively moderate sample sizes precluded specific analysis of subgroups.

A strength of this study is that these samples of transgender women of color living with HIV are from two of the largest studies with this specific population to date [46,47,48,49]. Further, the samples were from studies that were geographically diverse, drawn from three major regions of the US (West, East, and Midwest). Finally, we achieved consistent results in our validation analyses using two independent samples.

## 6. Future Directions

Development and initial validation of the PGA scale was conducted using samples from studies of transgender women living with HIV, the majority of whom were women of color. However, the PGA scale is designed to be applicable across populations of transgender women regardless of HIV status or race. Due to its potential ability to mitigate the negative health effects of transphobia, psychological gender affirmation is an important construct to study with people of all genders, regardless of HIV status [4,7,8,9,50]. Future research should explore applicability of the PGA scale in other groups of transgender and gender diverse people, such as transgender men and gender non-binary people, as well as in other geographic locations globally [51].

## 7. Conclusions

The PGA scale is a novel and psychometrically strong measure of psychological gender affirmation among transgender women. As a brief instrument, the PGA scale may be particularly useful in clinical and research settings to inform interventions to improve the health and well-being of transgender and gender diverse populations. The PGA scale can inform research that explores the impact of psychological gender affirmation on mental health, substance use, and other health outcomes among transgender women. In turn, research on psychological gender affirmation can inform gender-affirming clinical and social practices that are critical to preventing, addressing, and mitigating the downstream deleterious health effects of marginalization.

## Figures and Tables

**Table 1 ijerph-18-03298-t001:** Descriptive characteristics of participants in Study 1 (*N* = 278) and Study 2 (*N* = 368).

Characteristic	Study 1		Study 2	
Age in years-mean (std. dev)	43.5	(10.7)	34.2	(10.8)
	*N*	(%)	*N*	(%)
Race-Ethnicity				
Hispanic, Latina, or of Spanish origin	91	(32.7)	164	(44.6)
Black, non-Hispanic	126	(45.3)	187	(50.8)
White, non-Hispanic	19	(6.8)	-	-
Asian or Pacific Islander, non-Hispanic	8	(2.9)	1	(0.3)
Additional, non-Hispanic	3	(1.1)	1	(0.3)
Multiracial, non-Hispanic	30	(10.8)	5	(1.4)
No response	1	(0.4)	10	(2.7)
Education				
Less than grade 12	78	(28.1)	121	(32.9)
Grade 12	109	(39.2)	143	(38.9)
Some college or higher	91	(32.8)	87	(23.6)
No response	0	(0)	17	(4.6)
Financially secure ^1^	49	(17.6)	88	(23.9)
Experienced homelessness in previous 6 months	114	(41.0)	163	(44.3)
Sex work as a source of income in previous 6 months	50	(18.0)	134	(36.4)
Currently taking hormones	187	(67.3)	173	(47.0)

^1^ Study 1: ‘Currently’, Study 2: ‘in the previous 6 months’.

**Table 2 ijerph-18-03298-t002:** Standardized factor loadings from factor analyses.

Question Text	EFA Loading (Study 1)	CFA Loading (Study 2)	95% Confidence Interval of CFA Loading
	(*N* = 278)	(*N* = 368)	
How comfortable are you with going out in public during the day?	0.669	0.782	(0.736, 0.828)
How comfortable are you with people knowing that you are transgender?	0.635	0.730	(0.679, 0.781)
How satisfied are you with your body the way it is right now?	0.854	0.918	(0.899, 0.937)
How satisfied are you with the way you look right now?	0.899	0.975	(0.961, 0.989)
How satisfied are you with your current level of femininity?	0.732	0.843	(0.813, 0.873)

Notes: Exploratory factor analysis (EFA) factor loadings were estimated using FACTOR 10; confirmatory factor analysis (CFA) factor loadings and confidence intervals were estimated using Mplus 8.4.

**Table 3 ijerph-18-03298-t003:** Correlations of Psychological Gender Affirmation (PGA) with select study measures.

	Source	Correlation	95% Confidence Interval	*p*-Value
**Convergent Validity**				
Resilience	Study 1	0.300	(0.193, 0.407)	<0.001
Transgender Group Identification	Study 1	0.413	(0.315, 0.510)	<0.001
Ever on hormones	Study 2	0.113	(0.010, 0.216)	0.032
Currently on hormones	Study 2	0.117	(0.014, 0.220)	0.026
Body Satisfaction	Study 2	0.621	(0.557, 0.685)	<0.001
**Discriminant validity**				
Time since HIV diagnosis	Study 1	0.017	(−0.103, 0.136)	0.668
Experienced trauma	Study 1	−0.014	(−0.132, 0.103)	0.815
Ever experienced transphobia	Study 2	0.074	(−0.030, 0.179)	0.164
Education level	Study 2	0.071	(−0.034, 0.175)	0.184
Financial Insecurity	Study 2	−0.036	(−0.145, 0.072)	0.512

PGA: Scores on the Psychological Gender Affirmation scale. Sample size: 278 (Study 1), 368 (Study 2). Correlations estimated using full information maximum likelihood (FIML) in Mplus 8.4.

**Table 4 ijerph-18-03298-t004:** Bivariate associations of PGA with select outcomes.

Outcome	Source	Estimate (β)		*p*-Value
Positive affect	Study 1	1.256		0.001
Negative affect	Study 1	−0.913		0.002
		**Odds Ratio**	**95% Confidence Interval**	***p*-value**
Alcohol abuse/dependence	Study 1	0.662	(0.503, 0.871)	0.003
Illicit drug abuse/dependence	Study 1	0.737	(0.577, 0.942)	0.015
Depression ^1^	Study 2	0.729	(0.594, 0.895)	0.003

PGA: Scores on the Psychological Gender Affirmation scale Sample size: 278 (Study 1), 343 (Study 2; ^1^ Calculated for participants with non-missing scores for depression).

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
