# Peer review of "Measuring Satisfaction and Comfort with Gender Identity and Gender Expression among Transgender Women: Development and Validation of the Psychological Gender Affirmation Scale"

_ijerph, 2021, doi:10.3390/ijerph18063298_

Round 1
Reviewer 1 Report
Thank you for the opportunity to review your paper. It is an interesting work which can be enriched by more scientific support. Few points need consideration are as follows: 1. The introduction should be revised to establish a clearer and more compelling motivation for the study 2. The above points will also help in enriching the review of literature section. While the authors have tried to justify the proposed relationships in the model, they are missing a critical piece in their study, a theoretical foundation upon which they have built their model and study. Also, this does not provide justification for the study, variables, or other hypothesized relationships. 3. The results of discriminant validity, reliability and convergent validity test were not presented. (1) Significance level with t-value following CFA is suggested. (Also, AVE, CCR, and Cronbach’s alpha). Factor analysis (EFA and CFA) performed by variable does not make sense. (2) interval of plus or minus two standard errors around the correlation between the variables (3) compared to the free model using a χ2 difference test. (4) compared the AVEs for the two variables of interest with the square of the correlation between them (5) CMB related issues should be tested statistically with following suggestions of Podsakoff et al.(2003). Further to that it is very difficult to determine cause and effect relations in between variables in a cross-sectional design healthily. The point is the weakest part of the study.4. More scientific backing will help suggestions given for future research. 5. Finally, what are the practical implications based on findings of this study? More specific and realistic (substantial) implications are required. It is difficult to recognize difference from the already-preceded research. In addition, even the supplementation is necessary for research limitations and future research. Hope this will help in making the piece more interesting and applicable.Author Response
Reviewer 1
Thank you for the opportunity to review your paper. It is an interesting work which can be enriched by more scientific support. Few points need consideration are as follows:
- The introduction should be revised to establish a clearer and more compelling motivation for the study
Thank you for your interest in the motivation for the study. The motivation for the study is described throughout the introduction, as we describe the research gaps that currently exist in the understanding of psychological gender affirmation, despite evidence that psychological gender affirmation may provide a buffer against the negative health sequelae of transphobic experiences and prevent the internalization of transphobia. We also review the current lack of validated scales for measurement of psychological gender affirmation and describe how our study aims to address the gap in the research.
- The above points will also help in enriching the review of literature section. While the authors have tried to justify the proposed relationships in the model, they are missing a critical piece in their study, a theoretical foundation upon which they have built their model and study. Also, this does not provide justification for the study, variables, or other hypothesized relationships.
Thank you for the interest in the theoretical foundation of the study. The theoretical model upon which this study is based, the Model of Gender Affirmation, is first presented, cited, and described in paragraph 2 of the introduction. In paragraph 5 of the introduction, we explain how the current study is directly informed by the Model of Gender Affirmation, and in paragraph 6 we state that one of the primary aims of the paper was to “present evidence of the PGA scale’s relationship to variables that are hypothesized correlates as suggested by the Model of Gender Affirmation.” Taken together, these sections provide justification for the study as informed by our conceptual model as well as the hypothesized relationships between variables.
- The results of discriminant validity, reliability and convergent validity test were not presented.
The results of these tests are presented and can be found in the results section. The convergent and discriminant validity results are included in the subsection titled “Convergent and discriminant validity using Study 1 and Study 2 data” as well as in Table 3 titled “Correlations of PGA with select study measures”. Reliability information is included in the section titled “Confirmatory Factor Analysis of Study 2 data” where we state that “the reliability for the resulting 5-item scale was high (Cronbach’s alpha = .88, composite reliability = .93).” We now include the composite reliability in response to reviewer suggestion 3.1 below.
(3.1) Significance level with t-value following CFA is suggested. (Also, AVE, CCR, and Cronbach’s alpha). Factor analysis (EFA and CFA) performed by variable does not make sense.
Thank you for these suggestions. We now report the average variance extracted (AVE = .73) and also for composite reliability (CCR =.93). Please see the results section titled “Confirmatory Factor Analysis of Study 2 data”.
We have also revised the description of these analyses in the methods section, which now reads “As a measure of the internal consistency reliability, we calculated Cronbach’s alpha as well as composite reliability; values of 0.7 and greater are considered acceptable for both these measures. We also calculated average variance extracted (AVE) for which values above 0.5 are considered acceptable and those above 0.7 are considered very good.”
As stated above, the Cronbach’s alpha is included in the section titled “Confirmatory Factor Analysis of Study 2 data” where we state that “the reliability for the resulting 5-item scale was high (Cronbach’s alpha = .88, composite reliability = .93).”
We have also added the following citations for these analyses:
Werts, C.E., et al., A General Method of Estimating the Reliability of a Composite. Educational and Psychological Measurement, 1978. 38(4): p. 933-938.
Cronbach, L.J., Coefficient alpha and the internal structure of tests. Psychometrika, 1951. 16(3): p. 297-334.
Fornell, C. and D.F. Larcker, Evaluating Structural Equation Models with Unobservable Variables and Measurement Error. Journal of Marketing Research, 1981. 18(1): p. 39-50.
(3.2) interval of plus or minus two standard errors around the correlation between the variables
To find the confidence intervals provided for the correlations, please see Table 3 titled “Correlations of PGA with select study measures”.
(3.3) compared to the free model using a χ2 difference test.
Our analytic approach reflects our expectation that the scale will be used as an observed composite in applied research settings. Thus, the correlations were calculated with the composite scale scores and we do not have χ2 difference tests to report.
(3.4) compared the AVEs for the two variables of interest with the square of the correlation between them
Our approach has been to use the composite scores for the existing validated instruments as well as our new scale. This approach precludes the calculation of AVE for the two variables of interest.
(5) CMB related issues should be tested statistically with following suggestions of Podsakoff et al. (2003). Further to that it is very difficult to determine cause and effect relations in between variables in a cross-sectional design healthily. The point is the weakest part of the study.
We have a single factor scale. Upon a careful review the Podsakoff et al. (2003) paper, we examined the remedies that were applicable for our context and determined that we used all procedural remedies related to measurement design. We used rigorous methods to develop the scale, including cognitive interviewing and separation of the measures during survey administration. Further, none of the recommended optimal statistical adjustments outlined in that paper were appropriate for our study, which has a single-factor instrument.
- More scientific backing will help suggestions given for future research.
Thank you for this suggestion. The following sentence and citation were added to the Future Directions section to support suggestions provided for future research.
Lelutiu-Weinberger, C., D. English, and P. Sandanapitchai, The Roles of Gender Affirmation and Discrimination in the Resilience of Transgender Individuals in the US. Behavioral Medicine, 2020. 46(3-4): p. 175-188.
Fontanari, A.M.V., et al., Gender Affirmation Is Associated with Transgender and Gender Nonbinary Youth Mental Health Improvement. LGBT Health, 2020. 7(5): p. 237-247.
Hughto, J.M.W., et al., Social and Medical Gender Affirmation Experiences Are Inversely Associated with Mental Health Problems in a U.S. Non-Probability Sample of Transgender Adults. Archives of Sexual Behavior, 2020.
Goldenberg, T., et al., Stigma, Gender Affirmation, and Primary Healthcare Use Among Black Transgender Youth. Journal of Adolescent Health, 2019. 65(4): p. 483-490.
- Finally, what are the practical implications based on findings of this study? More specific and realistic (substantial) implications are required. It is difficult to recognize difference from the already-preceded research. In addition, even the supplementation is necessary for research limitations and future research. Hope this will help in making the piece more interesting and applicable.
Thank you for your interest in the practical implications of our study. This study is unique in that there are no existing validated measures of psychological gender affirmation, despite evidence that psychological gender affirmation may provide a buffer against the negative health sequelae of transphobic experiences and prevent the internalization of transphobia.
In the Conclusions section, we detail the implications of our study and potential uses of the validated scale:
“As a brief instrument, the PGA scale may be particularly useful in clinical and research settings to inform interventions to improve the health and well-being of transgender and gender diverse populations. The PGA scale can inform research that explores the impact of psychological gender affirmation on mental health, substance use, and other health outcomes among transgender women. In turn, research on psychological gender affirmation can inform gender-affirming clinical and social practices that are critical to preventing, addressing, and mitigating the downstream deleterious health effects of marginalization.”

Reviewer 2 Report
Dr.Sevelius and Dr. Neilands,
Thank you for your work on establishing the PGA scale for transgender women living with HIV.
Congratulations. The AHQR score is 9, which indicated that this research is of high quality.
|
item |
|
|
1) Define the source of information (survey, record review) |
Yes |
|
2) List inclusion and exclusion criteria for exposed and unexposed subjects (cases and controls) or refer to previous publications |
Yes |
|
3) Indicate time period used for identifying patients |
Yes |
|
4) Indicate whether or not subjects were consecutive if not population-based |
NA |
|
5) Indicate if evaluators of subjective components of study were masked to other aspects of the status of the participants |
NA |
|
6) Describe any assessments undertaken for quality assurance purposes (e.g., test/retest of primary outcome measurements) |
Yes |
|
7) Explain any patient exclusions from analysis |
Yes |
|
8) Describe how confounding was assessed and/or controlled. |
Yes |
|
9) If applicable, explain how missing data were handled in the analysis |
Yes |
|
10) Summarize patient response rates and completeness of data collection |
Yes |
|
11) Clarify what follow-up, if any, was expected and the percentage of patients for which incomplete data or follow-up was obtained |
Yes |
|
Score |
9 |
|
rate |
high |
1 As you mentioned (line 430), the study subjects are transgender women living with HIV. Could you explain to me that why you not included “HIV” in the style? I am looking for your insight.
2 For participants recruitment (line 131), could you draw a flowchart of participants inclusion with both studies? Figure 1 in the article below is an example.
Moreno, Jessica L., et al. "Agreement between self-reported psychoactive substance use and urine toxicology results for adults with opioid use disorder admitted to hospital." Toxicology communications 3.1 (2019): 94-101.
3 Could you cite articles to prove the studies have the largest samples (line 423-426)?
4 Could you list the data by region (West, East, and Midwest) in the supplementary/ appendix part(Line 426)?
Author Response
Reviewer 2
Thank you for your work on establishing the PGA scale for transgender women living with HIV.
Congratulations. The AHQR score is 9, which indicated that this research is of high quality.
Thank you for this positive feedback on our AHQR score.
|
item |
|
|
1) Define the source of information (survey, record review) |
Yes |
|
2) List inclusion and exclusion criteria for exposed and unexposed subjects (cases and controls) or refer to previous publications |
Yes |
|
3) Indicate time period used for identifying patients |
Yes |
|
4) Indicate whether or not subjects were consecutive if not population-based |
NA |
|
5) Indicate if evaluators of subjective components of study were masked to other aspects of the status of the participants |
NA |
|
6) Describe any assessments undertaken for quality assurance purposes (e.g., test/retest of primary outcome measurements) |
Yes |
|
7) Explain any patient exclusions from analysis |
Yes |
|
8) Describe how confounding was assessed and/or controlled. |
Yes |
|
9) If applicable, explain how missing data were handled in the analysis |
Yes |
|
10) Summarize patient response rates and completeness of data collection |
Yes |
|
11) Clarify what follow-up, if any, was expected and the percentage of patients for which incomplete data or follow-up was obtained |
Yes |
|
Score |
9 |
|
rate |
high |
1 As you mentioned (line 430), the study subjects are transgender women living with HIV. Could you explain to me that why you not included “HIV” in the style? I am looking for your insight.
Thank you for your interest in our study population. As stated in the discussion, development and initial validation of the PGA scale was conducted using samples from studies of transgender women living with HIV, the majority of whom were women of color. However, the scale is designed to be applicable across populations of transgender women regardless of HIV status or race.
2 For participants recruitment (line 131), could you draw a flowchart of participants inclusion with both studies? Figure 1 in the article below is an example.
Moreno, Jessica L., et al. "Agreement between self-reported psychoactive substance use and urine toxicology results for adults with opioid use disorder admitted to hospital." Toxicology communications 3.1 (2019): 94-101.
Thank you for this suggestion. A flowchart of participant inclusion is now included as “Supplementary Figure 1”.
3 Could you cite articles to prove the studies have the largest samples (line 423-426)?
Thank you for this suggestion. We have added the following recent citations from studies with transgender women that demonstrate that our studies include two of the largest samples of transgender women of color living with HIV to date.
Rosen, J.G., et al., Antiretroviral Treatment Interruptions Among Black and Latina Transgender Women Living with HIV: Characterizing Co-occurring, Multilevel Factors Using the Gender Affirmation Framework. AIDS and Behavior, 2019. 23(9): p. 2588-2599.
Reback, C.J., K.A. Kisler, and J.B. Fletcher, A Novel Adaptation of Peer Health Navigation and Contingency Management for Advancement Along the HIV Care Continuum Among Transgender Women of Color. AIDS and Behavior, 2019.
Mizuno, Y., et al., Factors Associated with Antiretroviral Therapy Adherence Among Transgender Women Receiving HIV Medical Care in the United States. LGBT Health, 2017. 4(3): p. 181-187.
Mizuno, Y., et al., Characteristics of Transgender Women Living with HIV Receiving Medical Care in the United States. LGBT Health, 2015. 2(3): p. 228-234.
4 Could you list the data by region (West, East, and Midwest) in the supplementary/ appendix part (Line 426)?
Thank you for this suggestion. The data is now disaggregated by region (West, East, and Midwest) as Supplementary Table 1: Descriptive characteristics of participants in Study 1 and Study 2 by geographic region of United States of America.

Round 2
Reviewer 1 Report
English editing is necessary.